# Monitoring of Cardiorespiratory Parameters during Sleep Using a Special Holder for the Accelerometer Sensor

**DOI:** 10.3390/s23115351

**Published:** 2023-06-05

**Authors:** Andrei Boiko, Maksym Gaiduk, Wilhelm Daniel Scherz, Andrea Gentili, Massimo Conti, Simone Orcioni, Natividad Martínez Madrid, Ralf Seepold

**Affiliations:** 1Ubiquitous Computing Lab, Department of Computer Science, HTWG Konstanz—University of Applied Sciences, 78462 Konstanz, Germany; maksym.gaiduk@htwg-konstanz.de (M.G.); wscherz@htwg-konstanz.de (W.D.S.); ralf.seepold@htwg-konstanz.de (R.S.); 2Dipartimento di Ingegneria dell’Informazione, Università Politecnica delle Marche, 60131 Ancona, Italy; andrea.gentili@staff.univpm.it (A.G.); m.conti@staff.univpm.it (M.C.); s.orcioni@staff.univpm.it (S.O.); 3School of Informatics, Reutlingen University, 72762 Reutlingen, Germany; natividad.martinez@reutlingen-unversity.de

**Keywords:** contactless measurement, accelerometer, health monitoring, ballistocardiography, sleep monitoring, heart rate, respiratory rate, vital signals

## Abstract

Sleep is extremely important for physical and mental health. Although polysomnography is an established approach in sleep analysis, it is quite intrusive and expensive. Consequently, developing a non-invasive and non-intrusive home sleep monitoring system with minimal influence on patients, that can reliably and accurately measure cardiorespiratory parameters, is of great interest. The aim of this study is to validate a non-invasive and unobtrusive cardiorespiratory parameter monitoring system based on an accelerometer sensor. This system includes a special holder to install the system under the bed mattress. The additional aim is to determine the optimum relative system position (in relation to the subject) at which the most accurate and precise values of measured parameters could be achieved. The data were collected from 23 subjects (13 males and 10 females). The obtained ballistocardiogram signal was sequentially processed using a sixth-order Butterworth bandpass filter and a moving average filter. As a result, an average error (compared to reference values) of 2.24 beats per minute for heart rate and 1.52 breaths per minute for respiratory rate was achieved, regardless of the subject’s sleep position. For males and females, the errors were 2.28 bpm and 2.19 bpm for heart rate and 1.41 rpm and 1.30 rpm for respiratory rate. We determined that placing the sensor and system at chest level is the preferred configuration for cardiorespiratory measurement. Further studies of the system’s performance in larger groups of subjects are required, despite the promising results of the current tests in healthy subjects.

## 1. Introduction

Ensuring healthy lives and promoting well-being for people of all ages is an important goal and agenda for society in the coming years [1,2]. This can be achieved by identifying interrelated strategic priorities such as promoting population health, achieving ubiquitous healthcare coverage and managing health emergencies. Statistics show that people spend up to 80% of their time in enclosed spaces, such as homes, offices and other buildings. In addition, we sleep at home for almost a third of our lives [3]. Poor sleep quality or disturbed sleep leads to daytime fatigue, which affects the mental and physical quality of daytime activities and increases the risk of accidents [4].

Sleep monitoring includes the monitoring of vital signs, which is particularly important given the increasing risk of cardiovascular disease [5]. This has resulted in a high prevalence of diseases that significantly impact quality of life. Therefore, the necessity and importance of monitoring vital sign parameters, including sleep quality, becomes extremely relevant.

Recently, polysomnography (PSG) has become the most widely accepted method of assessing sleep behavior, as it is the most accurate and leading sleep study approach [6]. This method involves, among others, monitoring a person’s brain activity, eye movements, heart rate and breathing during sleep [6,7]. This is usually carried out in a sleep laboratory. However, this approach is expensive, intrusive and time-consuming, making it impossible to do at home. This in turn does not allow continuous monitoring under these conditions. In this case, the development and use of a cost-effective system for outpatient or home use are of particular interest [8].

Several promising non-invasive, non-intrusive and inexpensive continuous home monitoring systems have been developed over the past decades [9,10]. On the other hand, recent advances in technology have led to the development of new approaches to measuring cardiorespiratory parameters during sleep using wearable devices to overcome the cumbersome use of PSG [11,12]. However, this approach is intrusive for the patient during sleep, which can affect the accuracy of the measurement results. At the same time, the rapid development of both signal processing methods and highly sensitive sensors has allowed attention to be focused on the method of recording cardiorespiratory parameters, such as ballistocardiography (BCG) [4]. BCG allows the non-contact measurement of heart rate and respiratory rate (RR) by studying the movement patterns propagating through an object mechanically connected to the subject [13]. For example, a bed can be used to track the heart rate of patients lying down overnight [14], resulting in a non-contact, unobtrusive measurement. Some non-invasive methods use ballistocardiography by placing sensors in the chair or bed where the patient is lying [15,16].

Researchers have proposed several ways to measure BCG signals during sleep. These methods include the use of different types of sensors, as well as differences in the number of sensors and their placement for measurement. Some of them used a hydraulic sensor system filled with water [17], load cells [18], piezoelectric load [19], pressure sensors installed under the mattress [20,21] and others. With the development of highly sensitive accelerometers based on micro-electromechanical systems (MEMS), their application in various measurement fields is growing rapidly [22,23,24,25]. Unfortunately, using accelerometers to measure BCG signals remains an understudied area of metrology [13,26,27].

This study aims to validate and test the developed reliable, unobtrusive and non-invasive system for cardiorespiratory measurements with a simple hardware setup convolution in a user-friendly way for heart rate and respiratory rate estimation. This system is based on an accelerometer sensor and includes a special holder setup for the sensor installation under the bed mattress. The additional aim of our study is to determine the optimum relative position (in relation to the subject or patient) of the system at which the most accurate and precise (in relation to existing ground truth systems) values of measured parameters could be achieved. These values are compared with the polysomnography reference measurement system to assess their accuracy. Moreover, in the study, we address the following questions:How can we use mechanical oscillations from the subject in cardiorespiratory measurements?Is there any complementary approach for cardiorespiratory monitoring using other non-invasive and non-contact technologies when achieving similar performance in relation to a resistive pressure transducer system [28,29]?

In Section 2 we present the proposed system with its setup in detail and signal processing algorithm. Section 3 details the obtained results. We highlight the results discussion and future work in Section 4 and conclude in Section 5.

## 2. Materials and Methods

Basically, the cardiorespiratory monitoring system consists of the following blocks: the mechanical part (including sensor holder and suspension) and data acquisition block. Each of these blocks is described in detail in the following subsections.

### 2.1. Mechanical Holder

The fact that measurement results are highly dependent on various aspects has been investigated in several studies [26,27,30]. The sensor holder is one of them because of the influence of the sensor vibrations under the mattress. It is important to mention that this holder should be installed in such a way that there is contact between the mattress and the sensor. The scheme of the system is shown in Figure 1.

Due to the lack of information on the characteristics of the holder, we have already presented the mechanical holder for respiratory rate measurement [27]. As mentioned above, spring steel (DIN EN 10 151 (Technical Information—DIN EN 10 151—https://www.din.de/de/mitwirken/normenausschuesse/fes/veroeffentlichungen/wdc-beuth:din21:52759011 accessed on 1 April 2023) was used as the material for the holder because of its ability to be practically unaffected by the weight of the mattress and to return to its initial state by respiratory movements and heart contractions. In this study, we used a 0.4 mm-thick holder with a 15 cm-long hanger, based on the previous results in the selection of acceptable holder parameters. Finally, Figure 2 and Figure 3 show the contactless system placed with the holder under the bed mattress and the structural scheme of the installation, respectively.

### 2.2. Data Acquisition

The signal acquisition system block consists of the computing unit that transmits data to the cloud or server, a storage module for data backup and a sensor. 

An ESP32 (Technical Information—ESP32—https://www.espressif.com/en/products/socs/esp32 accessed on 1 April 2023) (AZ-Delivery) module was chosen as the computing unit. One of the reasons for this choice is the size of this module: 54.5 × 27 × 11.5 mm^3^. These dimensions allow us to consider our system as a compact one. According to its specifications, this device can support voltages up to 5 V. In addition, the ESP32 module meets cost requirements. This unit sends data to a PC via Wi-Fi using an access point. In addition, a micro-SD module connected to the ESP32 allows each new acquisition to be stored locally as a backup copy of the data in case the Wi-Fi connection fails at the time it is sent to the PC.

In this paper, the ADXL355z (ADXL355z description and technical information—https://www.analog.com/media/en/technical-documentation/user-guides/eval-adxl354-355-ug-1030.pdf accessed on 1 April 2023) accelerometer (from Analog Devices) was used for data acquisition because of its good correlation between cost and accuracy of data acquisition. Its main features are low noise density, high sensitivity and programmable digital high- and lowpass filters. The data were recorded at a sampling frequency of 62 Hz, which is sufficient to obtain a cardiorespiratory signal. This sensor model has also previously been used in non-invasive measurements as part of studies on the performance of heartbeat and respiratory rate signal detection systems [13,26,31]. At the same time, this model has never been mentioned in these studies, which makes our study current.

We used SOMNO HD eco PSG (SOMNO medics GmbH, Randersacker, Germany) for obtaining the reference data. The respiratory, thorax (THO) and abdominal (ABD) and electrocardiogram (ECG) signals were recorded at sampling frequencies of 32 Hz and 256 Hz, respectively. 

### 2.3. Signal Processing and Analysis

The signal processing algorithm for heart rate and respiratory rate estimation consists of several steps. First, the first and last 10 s of data in each position were removed. It is necessary to avoid the signal epochs in which processing is complicated due to the presence of motion artefacts caused by the change in sleeping position. We then reduced the offset (isoline drift) from the raw data (or, in other words, compensation for gravitational acceleration) by subtracting the mean signal amplitude. In the next step, we derived the BCG and respiration signals from the accelerometer sensor using the Butterworth bandpass filters. For the BCG signal, the filter has a 6th order in the range [3; 15] Hz, and for the respiration signal a 4th order in the range [0.15; 0.4] Hz. The processing algorithms for the two signals are also different. For this reason, we will first describe the action sequence for the BCG signal and then for the respiratory signal. 

#### 2.3.1. HR Estimation Algorithm

Considering the multicomponent nature of physiological signals, we must take into account all 3 existing sensor axes [22]. In our case, the *x-axis* detects side-to-side vibrations of cardiac activity. The *y-axis* that corresponds to vibrations in the dorsoventral direction is perhaps the most commonly used seismocardiography (SCG) signal axis, and it is thought to contain information on various phases of the cardiac cycle such as the mitral valve closure, isovolumetric contraction, aortic valve opening rapid ejection, aortic valve closure, mitral valve opening and rapid filling. Finally, the *z-axis* senses vibrations in the head-to-foot direction [22,32]. It is important to note that combining information from multiple axes of the signal from an accelerometer in cardiorespiratory measurements can result in improved hemodynamic monitoring compared to when single axes are used [32]. Moreover, the combination of all axes allows vibrations to be detected in the head-to-foot and dorsoventral directions coupling to the *x-axis* signals due to imperfect sensor axis-to-body axis alignment [33]. Therefore, it is necessary to calculate the magnitude of obtained BCG signal in the next step. The signal magnitude is calculated by using the formula:(1)$$A=\sqrt{x^{2}+y^{2}+z^{2}},$$
where *x*, *y* and *z* are the current values of sensor amplitude.

After that, we applied the 6th-order Butterworth bandpass filter in the range [0.7; 3.25] Hz to limit heart rate detection between 42 bpm and 195 bpm. In the final stage, the moving average filter was applied to the signal to extract signal peaks relevant to R-peaks on the ECG signal. HR estimation was conducted by the peak detection function applied to the obtained and processed signal.

#### 2.3.2. RR Estimation Algorithm

It is important to minimize the detection error of false breath peaks. Because of that, we applied the moving average filter for the signal of each accelerometer sensor axis. After that, as a final signal processing stage, the signal magnitude was calculated based on the same theoretical aspects as the BCG signal. RR estimation, like HR, was conducted by the peak detection function applied to the obtained and processed signal in MATLAB.

A block diagram of the BCG and respiratory signal processing algorithm is shown in Figure 4 to illustrate the sequence of operations.

### 2.4. Experiment Design

For our investigation, we used a regular single bed (Askvoll), bed net/slatted (Lüroy), and mattress with a dimension of 90 cm × 200 cm from IKEA (IKEA, Delft, The Netherlands). All materials were wooden and widely accessible by regular users.

It is important to notice that we acquired the data from 4 sensor positions during the measurements to determine a more appropriate sensor position in relation to the subject. In particular, these positions correspond to the level of the chest (S1), xiphoid process or, in other words, solar plexus (S2), diaphragm (S3) and abdominal muscles (S4). The scheme of considering sensor positions is presented in Figure 5.

Before the experiment, the subjects were instructed to lie down on the bed in four regular sleeping positions: prone (P1), right lateral (P2), supine (P3) and left lateral (P4) (see Figure 6). The experiment started on the prone and ended on the left lateral position in a counterclockwise rotation of the subjects. All subjects had at least five minutes as relaxing time before data collection. The data measurement lasted 140 s in each position. The subjects were instructed to behave normally with minimum movement during the data collection. However, they were informed that in case of any inconvenience, the experiment would be stopped. The measurements were repeated 4 times due to the necessity of data collection in relation to mentioned sensor positions. 

## 3. Results

### 3.1. Subjects’ Statistical Data

The data were collected from 23 subjects in our study. In particular, we engaged to take part in the experiment for 13 males and 10 females with an average age, height and weight of 31.4 ± 7.1 years, 173.0 ± 7.3 cm and 73.2 ± 10.5 kg. All subjects were informed of the consent form before the experiment. The subjects involved in the study were informed regarding the specifics of the research. The subjects did not have or mention any known pulmonary, cardiovascular or other types of disorders and diseases and were not receiving any kind of treatment or medication. Table 1 shows the statistical data of recruited subjects in our study. 

The determination of sensor position for the contactless system using an accelerometer was conducted by heart rate (HR) estimation and respiratory rate (RR) separately in comparison with reference data. The mean average error (MAE) was chosen as an evaluation metric in this research. It is important to mention that each signal epoch was separated into 20 s chunks for further HR and RR analysis and MAE calculation. Based on the experiment design and study details presented in previous sections and subsections, we considered about 552,20 s signal windows for each sensor position for further analysis (138 for each sleeping posture). Therefore, we analyzed 2208 signal windows with a total length of 52,440 s (around 14.5 h) in this research.

### 3.2. Heart Rate Monitoring

The first parameter is the heart rate (HR) for the performance evaluation of the developed system. The analysis was conducted with the abovementioned number of signal segments. Table 2 shows the values of MAE, which can help us to estimate a preferable sensor position of the accelerometer sensor for HR measurement.

In addition to this result, we separated the results into two blocks by gender not only for sensor position detection by this sign but also for evaluating the peculiarities in HR estimation between males and females by our developed system. Thus, we obtained 1248 signal windows for males (312 for each sensor position; 312 for each sleeping posture) and 960 signal segments for females (240 for each sensor position and 240 for each sleeping posture). Table 3 provides these results for females and males, respectively.

Figure 7 shows an example of an analyzed subject’s BCG signal compared the reference ECG signal. The processed accelerometer signal is on the top, and the signal obtained by the polysomnography reference system is on the bottom.

The results presented in Table 2 and Table 3 allow us to notice several features for determining the most appropriate sensor position in relation to the patient for HR estimation. Therefore, the most accurate results of heart rate could be delivered from the S1 (chest level) sensor position. This is confirmed by the minimal of the calculated average values of MAE for all sleeping postures (2.24 bpm) and in each separately considered sleeping posture. It could be noticed that the results related to S1 in P1 (prone) and P3 (supine) postures correspond regarding the HR detection with approximately the same MAE values (2.15 bpm and 2.29 bpm, respectively). The other sensor positions could obtain the HR with a greater difference in relation to P1 and P3 MAE values and larger values of MAE in general. The less interesting point is the minimal value of MAE of heart rate detection in P2 posture among the considered postures for S1 (2.13 bpm). This allows us to declare the HR estimation almost without losing precision results, at least in these sleeping postures. However, the difference between, for instance, prone (P1) and right lateral (P2) postures is not significant (0.02 bpm). At the same time, the values related to the S2 (solar plexus level) sensor position also present acceptable results (MAE is less than 3 bpm) in all considered sleeping postures.

It must be noted that the HR estimation is almost equal to precision in lateral directions P2 (right lateral side) and P4 (left lateral side) for each sensor position S_i_. At the same time, for S1 and S2, the heart rate in the right lateral side (P2) is detected better than in left lateral side (P4) for S1 and S2 sensor positions. This is perhaps caused by more precise installation of the electrodes of the reference system (in turn, it could be influenced by the results) or by physiological aspects of the vibrations expansion from heart contractions. The other interesting point from the obtained results is the lower MAE value for females than for males. For example, the average MAE value in S1 is 2.19 bpm against 2.28 bpm for males. However, the results are almost identical for both genders. We noted that occasional movements during the experiment were more frequently observed in males, which is a direct cause of the presence of signal artefacts.

In addition to the presented results regarding S1, we discovered that the trend of the change in the detected heart rate has a clear correlation with the changes in the reference data. This pattern is not observed in all subjects but is presented in a significant number of data. The discrepancy with the trend in all datasets is explained by the presence of possible signal artefacts, the minimization of which is not currently implemented in the processing algorithm. Figure 8 shows this for two subjects (male and female) in the S1 sensor position.

Based on the described results, we can point out that the sensor position at the chest level (S1) is the most appropriate position for the HR estimation for the developed system with an accelerometer sensor.

### 3.3. Respiratory Rate Monitoring

The second parameter of vital signs which we can detect by using our system is respiratory rate. The analysis was conducted using MAE estimation for the same number of 20 s signal windows. Table 4 shows the results for all subjects in comparison with thorax (THO) respiration and with abdominal (ABD) respiration belts. 

In addition to that, we split the results for females and males to assess the details of respiration depending on gender. Table 5 relates to the results for thoracic respiration and the comparison between females and males. Table 6 relates to the results for abdominal respiration and the comparison between females and males.

Figure 9 shows an example of an analyzed subject’s respiratory signal compared to reference signals obtained from respiratory belts. The processed respiratory signal from the ADXL355 sensor is on the top, the thorax respiratory signal is in the middle and the abdominal respiratory signal obtained by the polysomnography reference system is on the bottom.

Based on the results presented in Table 4, we notice that the S1 sensor position (chest level) allows us to obtain the most precise results for respiration rate estimation. Moreover, these results are related to both respiration signals, considered from the reference system (1.51 rpm for thoracic and 1.31 rpm for abdominal). The additional interesting point is that S2 (solar plexus level) and S3 (diaphragm level) delivered similar average results of MAE for both respiration kinds. At the same time, this trend was not noticed during the HR results analysis. Thus, MAE values for the S2 sensor position are 1.67 rpm for THO and 1.47 rpm for ABD and 1.69 rpm and 1.50 rpm for the S3 sensor position, respectively.

In general, the obtained results correspond to abdominal respiration being detected more accurately than thorax. This is possibly confirmed by practical information from previous research [34]. However, theoretically, these signals have almost the same RR values and the difference is only in signal amplitude. In our case, we can explain varying THO and ABD values by the possible predominance of one type of breathing over the other in some subjects. However, it is not the purpose of our study to determine the predominant type of breathing.

P1 (prone) sleeping posture allows us to extract RR with smaller MAE values (1.19 rpm for THO and 1.04 rpm for ABD) as well as HR in the same sensor position (S1). In this case, the MAE values in lateral sleeping postures (P2, P4) are worse in abdominal respiration (1.41 rpm and 1.42 rpm, respectively) than in frontal direction positions (P1–1.04 rpm, P3–1.37 rpm). At the same time, a slight opposite trend is noticed for thoracic respiration. Therefore, the P2 (right lateral) and P3 (supine) positions are detected with worse MAE values (1.68 rpm and 1.62 rpm, respectively) than in the two other sleeping postures.

Based on the results presented in Table 5 and Table 6, the RR values estimated from males are better than from females in both respiration kinds. The other point is that the S3 sensor position is better than S2 for males in respiratory rate estimation for the abdominal. Thus, the average values for ABD in S3 are 1.53 rpm (females) and 1.48 rpm (males) and in the S2 sensor position 1.61 rpm and 1.59 rpm, respectively.

Thus, based on the described results, we can point out again that the sensor position at the chest level (S1) is the most appropriate position for the RR estimation for the developed system with an accelerometer sensor. However, the S2 sensor position (solar plexus) could be considered an acceptable position for system installation or as an opportunity for two-sensor installation in parallel in case the first sensor breaks down. 

## 4. Discussion

### 4.1. Remaining Challenges

Despite the obvious application of the accelerometer sensor in ballistocardiography (BCG) to measure HR and RR, the objective of contactless detection of these vital sign parameters in sleep with high accuracy is still challenging for several reasons. Some of the reasons are signal-to-noise ratio, signal morphology (especially when calculating the signal in all directions) and sufficiently high sensor sensitivity to different oscillations in an environment that can be affected by, for instance, changes in the subject’s body position. In general, the idea of the system development, which could measure HR and RR in different sleeping postures on a reliable level, is well known. The system should have sufficiently high accuracy and be easy to use. In the case of accelerometer sensor application to this system, the problem of the region of interest for sensor installation is a significant challenge as well as the sensor position in relation to the patient. Therefore, we can detect the vital signs parameters with significant accuracy only when the subject position is not on the bed edge. This problem could be partially solved by installing two accelerometer sensors in different parts of the bed [35]. However, the authors detected HR and RR when the subject was in a sitting position. The solution of covering most of the bed area was solved by using other types of sensors, such as quartz resonator (QCR), force-sensing resistor sensor (FSR) or fiber-optic [29,36,37]. However, this research is related to the usage of other technologies (not based on the accelerometer sensor). In most of the considered cases, the use of mentioned technologies increases the complexity of the system, its maintenance and costs.

### 4.2. Enhancements, Applications and Features

During the study, we determined the position of the sensor in relation to the patient that gives the most accurate HR and RR values. This sensor position is at chest level. However, we also allowed the sensor to be placed at the level of the solar plexus if, for example, the bed construction did not allow the system to be placed at the aforementioned level. In turn, for the most acceptable sensor position, we were able to achieve a difference in HR estimation error (depending on sleep position) of 0.06 bpm and 0.07 bpm for males and females, respectively, and in RR estimation of 0.10 rpm and 0.05 rpm for males and females, respectively, for thoracic breathing and 0.01 rpm for males and 0.06 rpm for females for abdominal breathing. Such indicators may be important for clinical trials and sleep monitoring systems [38], as they do not require additional measures and conditions to be imposed on the patient during diagnosis. In addition, such a system with minimal reliance on real-life sleep position, where there is minimal control over the subject, may have benefits in terms of obtaining longer periods of reliable signal, as the subject does not need to follow specific instructions (or at least less so).

During a system check procedure before the start of the experiment, we pointed out the possible potential in determining sleep stages [39]. This could be achieved by implementing a heart rate variability algorithm. Thus, this system has the potential to be not only an HR and RR monitor, but also an attractive addition to wearable devices that are quite intrusive [22,24,40]. Another opportunity for system enhancement is the implementation of different algorithms for sleep disorder detection. One of them could be the algorithm for sleep apnea detection [41].

Of course, the signal-to-noise ratio affects the quality of estimating of the parameters of the vital signals, as mentioned above. One of the possible improvements to this characteristic is the use of a dome-shaped addition to the sensor [42,43]. This dome has been shown earlier to act as a signal amplifier for a resistive pressure sensor in cardiorespiratory measurements. At the same time, it is necessary to conduct further research into the size of the dome, the material that can maximize the signal and the location of the installation of such a dome. These studies are the next step in our work to improve the system’s quality.

It is important to note that the accelerometer can introduce some systematic errors, even under identical experimental conditions. This can affect the stability and reproducibility of the signal and data. However, we tried to eliminate the influence of external factors by maintaining the same experimental conditions for all subjects. This includes:Keeping the environment isolated from sound pollutants (a possible reason for different oscillations);Air conditions such as temperature, humidity and pressure by continuously measuring and monitoring these parameters;Conducting experiments at fixed times of the day with some tolerance;Leaving the bed unoccupied for some time (duration is up to 10–15 min) to restore the potential drift of the sensors to the initial states from previous experiments (influenced by the weight, force and surface of objects).

During the experiment, we did not observe any particular discrepancy in the readings or any drop in system performance. Looking at the subjects’ weight values, we found no relationship between the MAE values before or after a subject whose weight was significantly different from the others. The situation is similar for floors. We ran the experiments in a completely random order of gender, and the results did not take into account the order of the experiment.

The other aspect we should note is the range of HR and RR values in the data obtained. Thus, we covered a wide range of HRs from 46 to 138 and RRs from 9 to 24, and the system still needs to be evaluated with subjects outside this range. This could be solved, for example, by enrolling subjects from broader age groups and anthropometric conditions. It is also possible to simulate the change in HR and RR during the experiment by simulating physical activity before the measurement (which should increase heart rate and respiratory rate for several minutes). However, in this case we must reduce the relaxation time before signal recording and collect the data from the subject separately in each sleeping position, as the subject rests (and recovers) during each measurement.

One of the main goals of this work is to develop a reliable, non-intrusive and non-invasive system for cardiorespiratory measurements. Therefore, our approach must meet the stated criteria. At the same time, as noted in previous studies, it is equally difficult to meet all the stated criteria because the priority of the criteria differs according to the purpose of the systems [29,44]. Therefore, the analysis of our system and the comparison with previously presented devices is reduced to a better comparison. Table 7 shows previously developed cardiorespiratory monitoring systems that meet the stated criteria in comparison with our system.

To compare the systems presented above, we defined unobtrusiveness as the ability not to create additional conditions for patients in the monitoring process. In addition, the ability to keep data confidential was taken into account. For example, systems based on cameras [45,46] cannot be considered unobtrusive. Using fiber-optic technologies in cardiorespiratory monitoring can provide the most accurate results compared to [47,48] reference systems. However, the cost of such technologies is quite high, and, combined with the complexity of maintaining such a system, the widespread use of such devices may not be feasible. On the other hand, the implementation of systems based on resistive pressure sensors and accelerometers (including our system) involves unobtrusive monitoring (sensors are installed under the bed mattress). In addition, the systems presented are very easy to maintain and offer the possibility of quickly replacing each module in the event of failure. At the same time, the placement of sensors with the greatest coverage of the region of interest in cardiorespiratory monitoring showed promising results in terms of accuracy [29]. Installing the accelerometer under the sofa, as well as from one of the edges, makes it possible to obtain acceptable results in determining respiratory rate but not in heart rate accuracy [35]. Therefore, the results obtained during the study allow us to state that the system developed meets the criteria mentioned. 

Regarding the cost of the system, we did not include any information in the table above. This is mainly due to the lack of information from the authors of the cited works on the cost of producing the prototype. At the same time, we can state that the indicative cost of our system is between EUR 85 and 100, depending on the cost of the components of the system holder.

### 4.3. Limitations

It should be noted that we measured healthy subjects with regular cardiorespiratory parameters. Patients with irregular parameters may require a change in bandwidth frequency. Deep diaphragmatic or thoracic breathing may affect the final system assessment, which requires further investigation. In addition, we were able to recruit subjects with similar anthropometric parameters. Therefore, subjects with different weights and heights should be considered for further investigation.

## 5. Conclusions

We tested and validated the ADXL355z accelerometer sensor in combination with the developed mechanical sensor holder to address issues of system reliability and accuracy, as well as to reduce the dependence of sensor sensitivity on the subject’s position during sleep. This system can be considered as simple yet inexpensive to design using a single sensor and a minimum number of modules. This system could be used in a variety of ways, such as at home, in outpatient clinics or care units. This study has highlighted the detection of small heart and respiratory activity, and therefore estimation of heart and respiratory rate could be achieved by increasing the sensitivity of the accelerometer sensor. This in turn allows for a simplified holder design and subsequent implementation of the signal processing algorithm.

We determined that the sensor position at chest level gave the most acceptable and accurate results (with the lower value of MAE) for cardiorespiratory measurements. Using the aforementioned sensor, we were able to achieve the reliability and accuracy of the system to error values of 2.24 bpm for HR and 1.51 rpm and 1.31 rpm for thoracic and abdominal respiration, respectively, in the defined best sensor position. The mean MAE of RR estimation was 1.30 rpm (for thoracic) and 1.41 rpm (for abdominal) for females and 1.21 rpm and 1.39 rpm for males for THO and ABD respiration, respectively. For HR estimation, the results were 2.19 bpm for females and 2.28 bpm for males. In addition, the trend of the change in the detected heart rate value of the developed system coincides with the trend of the difference in the reference data, which indicates the reliability of the determined values and the reliability of the presented results.

## Figures and Tables

**Figure 1 sensors-23-05351-f001:**
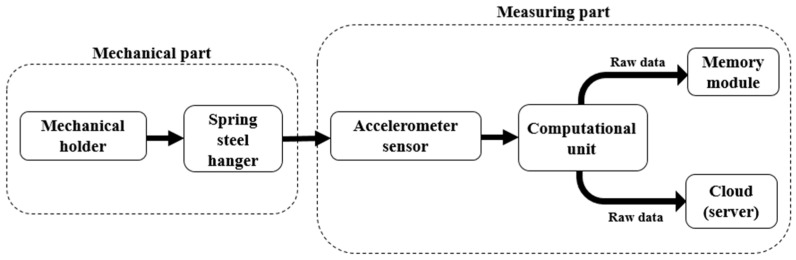
The structure scheme of the system.

**Figure 2 sensors-23-05351-f002:**
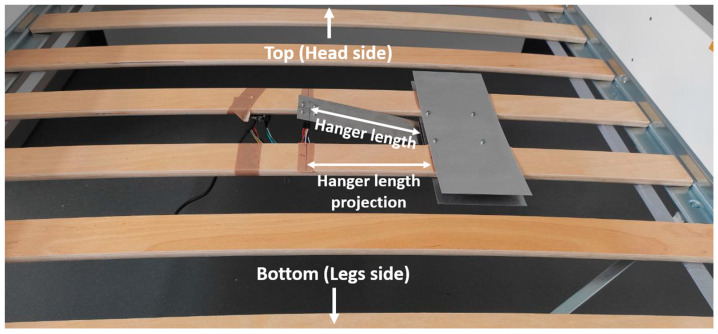
The contactless system for cardiorespiratory monitoring on the bedframe.

**Figure 3 sensors-23-05351-f003:**
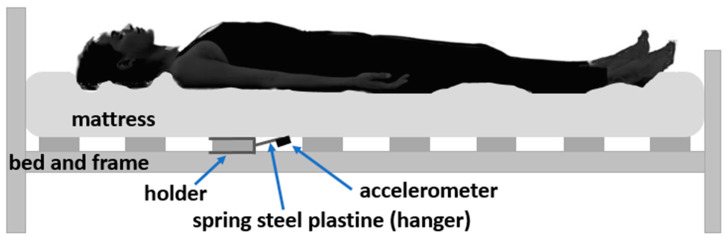
The structure scheme of system installation.

**Figure 4 sensors-23-05351-f004:**
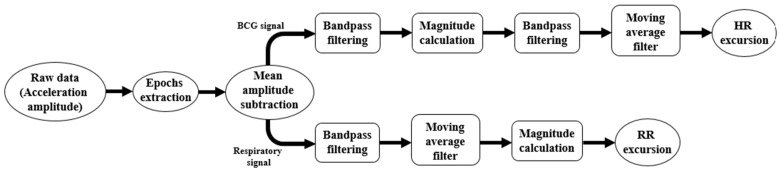
The signal processing algorithm block diagram.

**Figure 5 sensors-23-05351-f005:**
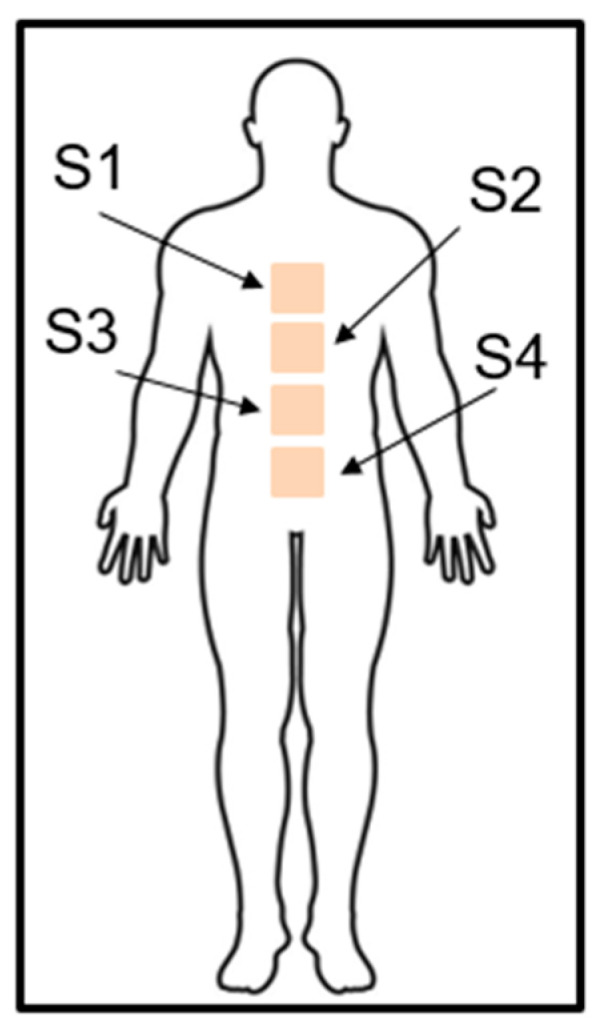
The scheme of considering sensor distribution.

**Figure 6 sensors-23-05351-f006:**
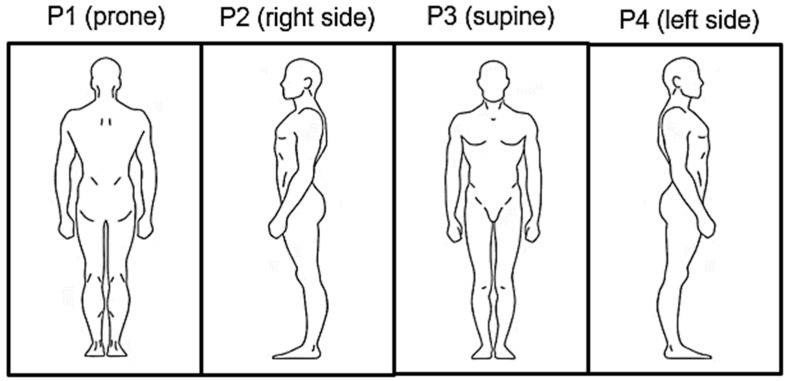
The sequence of the subject’s positions during the experiment on the bed.

**Figure 7 sensors-23-05351-f007:**
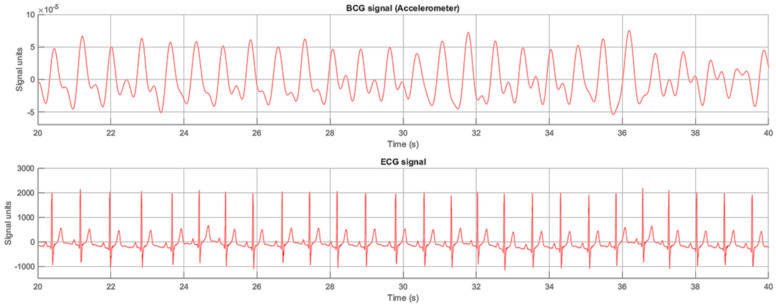
Example of processed BCG signal from ADXL355z and ECG signal from PSG system (Subject 10).

**Figure 8 sensors-23-05351-f008:**
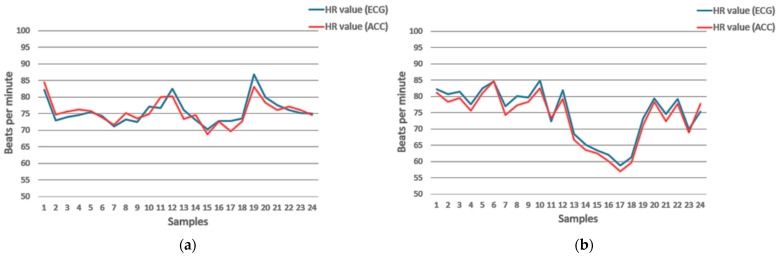
Example of changing HR values from our (red line) and reference (blue line) system. (**a**) The data from a male subject (Subject 1); (**b**) the data from a female subject (Subject 18).

**Figure 9 sensors-23-05351-f009:**
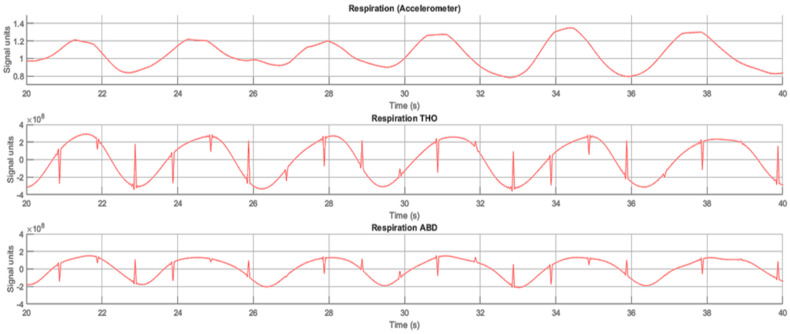
Example of processed respiratory signal from ADXL355z and thoracic and abdominal respiratory signals (RIP Tho, RIP Abd) from the PSG system (Subject 10).

**Table 1 sensors-23-05351-t001:** Statistics of recruited subjects in this study.

Subject	Gender	Age (Years)	Height (cm)	Weight (kg)
1	Male	26	179	72
2	Male	25	181	74
3	Male	40	170	65
4	Female	25	168	67
5	Female	37	160	52
6	Male	36	171	80
7	Female	33	167	73
8	Male	31	177	70
9	Female	24	155	50
10	Female	43	178	78
11	Male	23	179	65
12	Male	34	184	75
13	Male	34	180	67
14	Male	28	179	88
15	Female	23	175	62
16	Male	53	176	82
17	Female	24	171	87
18	Female	19	169	69
19	Female	27	150	60
20	Male	54	188	85
21	Female	25	166	60
22	Male	32	179	130
23	Male	27	178	72

**Table 2 sensors-23-05351-t002:** The heart rate estimation of the accelerometer sensor by MAE (bpm).

Subject Position	Sensor Position			
	S1	S2	S3	S4
Prone (P1)	2.15	2.34	2.75	2.51
Right lateral (P2)	2.13	2.37	3.12	3.37
Supine (P3)	2.29	2.92	2.80	3.40
Left lateral (P4)	2.40	2.58	3.00	3.21
Average	2.24	2.53	2.92	3.12

**Table 3 sensors-23-05351-t003:** The heart rate estimation (by gender) of the accelerometer sensor by MAE (bpm).

Subject Position	Sensor Position							
	S1		S2		S3		S4	
	Males	Females	Males	Females	Males	Females	Males	Females
Prone (P1)	2.14	2.18	2.29	2.39	2.33	3.29	2.49	2.53
Right lateral (P2)	2.20	2.04	2.28	2.49	2.77	3.61	3.14	3.65
Supine (P3)	2.33	2.24	2.92	2.65	3.05	2.49	3.20	3.67
Left lateral (P4)	2.47	2.31	2.43	2.78	2.77	3.27	3.35	3.04
Average	2.28	2.19	2.50	2.57	2.70	3.19	3.06	3.21

**Table 4 sensors-23-05351-t004:** The respiratory rate estimation of the accelerometer sensor by MAE (rpm) compared with thoracic and abdominal respiration belts.

Subject Position	Sensor Position							
	S1		S2		S3		S4	
	THO	ABD	THO	ABD	THO	ABD	THO	ABD
Prone (P1)	1.19	1.04	1.48	1.36	1.49	1.30	2.18	1.91
Right lateral (P2)	1.68	1.41	1.61	1.33	1.72	1.48	1.79	1.62
Supine (P3)	1.62	1.37	1.65	1.48	1.88	1.63	1.92	1.84
Left lateral (P4)	1.57	1.42	1.95	1.72	1.66	1.61	1.99	1.69
Average	1.51	1.31	1.67	1.47	1.69	1.50	1.97	1.77

**Table 5 sensors-23-05351-t005:** The respiratory rate estimation of the accelerometer sensor by MAE (rpm) compared with thoracic respiration belt.

Subject Position	Sensor Position							
	S1		S2		S3		S4	
	Males	Females	Males	Females	Males	Females	Males	Females
Prone (P1)	1.05	1.02	1.29	1.18	1.44	1.31	1.91	1.95
Right lateral (P2)	1.16	1.37	1.06	1.27	1.43	1.50	1.44	1.26
Supine (P3)	1.26	1.32	1.40	1.42	1.62	1.79	1.70	1.81
Left lateral (P4)	1.36	1.51	1.55	1.77	1.67	1.55	1.42	1.50
Average	1.21	1.30	1.32	1.41	1.54	1.54	1.62	1.63

**Table 6 sensors-23-05351-t006:** The respiratory rate estimation of the accelerometer sensor by MAE (rpm) compared with the abdominal respiration belt.

Subject Position	Sensor Position							
	S1		S2		S3		S4	
	Males	Females	Males	Females	Males	Females	Males	Females
Prone (P1)	1.03	1.05	1.41	1.45	1.20	1.35	1.91	2.10
Right lateral (P2)	1.61	1.65	1.54	1.60	1.52	1.63	1.77	1.94
Supine (P3)	1.46	1.59	1.54	1.57	1.63	1.67	1.94	1.74
Left lateral (P4)	1.46	1.36	1.85	1.82	1.57	1.48	1.90	2.09
Average	1.39	1.41	1.59	1.61	1.48	1.53	1.88	1.97

**Table 7 sensors-23-05351-t007:** Comparison of this work to the current state-of-the-art.

	Number of Subjects	Sensor Type	Reliability				Unobtrusiveness
			HR MAE, bpm	RR MAE, rpm	HR MAPE, %	RR MAPE, %	Yes or No
[45]	5	Camera	7.40	-	12.46	-	No
[46]	40	Camera	2.80	2.10	-	-	No
[47]	10	Fiber optic	2.00	1.00	-	-	Yes
[48]	10	Fiber optic	-	-	5.41	11.60	Yes
[29]	20	FSR	3.24	2.32	-	-	Yes
[35]	45	ACC	10.10	1.76	3.60	6.25	Yes
This work	23	ACC	2.24	1.52	-	-	Yes

## Data Availability

The data presented in this study are available on request from the corresponding author.

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
