# Peer review of "Monitoring of Cardiorespiratory Parameters during Sleep Using a Special Holder for the Accelerometer Sensor"

_sensors, 2023, doi:10.3390/s23115351_

Round 1

Reviewer 1 Report

This article provides a very detailed review of methods performed, results, discussion and conclusions related to the use of under mattress accelerometery in detecting sleep behaviors. This detailed discussion provides a very clear and cogent analysis of the benefits and future directions needed to validate this process.  Minor grammatical/language errors could be corrected.

Minor grammatical/language errors could be corrected.

Author Response

Dear Sir/Madam,

Please find in the attachement the revised version of article based on the comments and suggestions from you and your colleagues.

Best regards,

Andrei Boiko

Reviewer 2 Report

The manuscript presented a novel system based on MEMS accelerometer for monitoring cardiorespiratory parameters. The research was well designed and properly presented. Please consider improving the paper from the below aspects:

  1. Line 65, please spell out the abbreviation “RR” the first time it appears in the manuscript.
  2. Line 70, please rephrase the sentence “They 70 use different types of sensors, their number and placement.” 
  3. Line 74, MEMS stands for “Micro-electromechanical systems”.
  4. Line 75, the authors mentions “using accelerometers to measure BCG signals remains an understudied area”, could you provide references for latest research?
  5. Line 77, please rephrase “…and test the developed reliable a reliable,…”
  6. Table 7, the ref. [37] is listed without any reliability results. Is this purposeful?
  7. Same in Table 7, can the authors add a column to compare the cost and/or complexity of the exiting works?

Minor changes are required.

Author Response

Dear Sir/Madam,

Please find in the attachement the revised version of article based on the comments and suggestions from you and your colleagues.

  1. Comments no. 1-5 were revised and corrected.
  2. We have provide the description related to the costs for the compared systems in the Table 7 after this table. Additionally, we provided the information regarding our costs for the producing the prototype.

Best regards,

Andrei Boiko

Reviewer 3 Report

The paper under review proposes a non-invasive unobtrusive cardiorespiratory parameter monitoring system based on an accelerometer sensor. The general subject is of great interest in bio-sensing and the paper gives a valuable contribution. The paper is written clearly, but the quality of composition could be improved after a careful proofreading. Here are some cases that the authors might be willing to rewrite (more instances could be caught up by authors:

1. line 15:  Maybe you want to rephrase "Consequently, developing a non-invasive and non-intrusive home sleep monitoring system that can reliably and accurately measure cardiorespiratory parameters while causing minimal influence on patients is of great interest" as "Consequently, developing a non-invasive and non-intrusive home sleep monitoring system with minimal influence on patients, that can reliably and accurately measure cardiorespiratory parameters, is of great interest",

2. line 38: "According to statistics, people spend up to 80% of their time in enclosed spaces such as the home". This is too general and possibly not a valid claim (in this generality), and needs to be rephrased,

3.  line 39: "In this turn we spend almost a third of our lives sleeping there" (!?). Needs to be rewritten, also the reference seems inappropriate,

4. line 43: "The latter is particularly important given the increasing risk of cardiovascular disease". Vague and too fast a conclusion,

5. line 50, "among other things". Add a comma here,

6. line 56: "efforts have been made to..." This could be rephrased better,

7. line 57: "In turn" (!?) You mean "Accordingly" or "On the other hand"?

8. line 60-61: "leaves an element of" and "the measurement result" (?) Could be rephrased,

9. line 70: "the developed reliable a reliable" (!?),

10. line 92: "presented" should be "present",

11. line 96: "Considered as a basis" (?) Not such a good start for a paragraph,

12. there are much more, but I stop here, hoping that the authors do a careful proofreading,

13. References need to follow a consistent style: for instance using all "capital letters" or not, etc.  

I only sketched a few examples of needed editing. You need to take care of the whole paper (possibly with help from an expert).  

Author Response

Dear Sir/Madam,

Thank you very much for your comments and suggestions related to the article!

Please find in the attachement the revised version of article based on the comments and suggestions from you and your colleagues.

All comments and suggestions were considered and taken into account during the revision procedure. In addition to that, we have revised English throughout the paper.

Best regards,

Andrei Boiko
